# Nanoparticle Vaccines for Inducing HIV-1 Neutralizing Antibodies

**DOI:** 10.3390/vaccines7030076

**Published:** 2019-07-29

**Authors:** Mitch Brinkkemper, Kwinten Sliepen

**Affiliations:** Department of Medical Microbiology, Amsterdam Infection & Immunity Institute, Amsterdam UMC, University of Amsterdam, Meibergdreef 9, 1105AZ Amsterdam, The Netherlands

**Keywords:** HIV-1 Env, virus-like particles, liposomes, self-assembling protein nanoparticles

## Abstract

The enormous sequence diversity between human immunodeficiency virus type 1 (HIV-1) strains poses a major roadblock for generating a broadly protective vaccine. Many experimental HIV-1 vaccine efforts are therefore aimed at eliciting broadly neutralizing antibodies (bNAbs) that are capable of neutralizing the majority of circulating HIV-1 strains. The envelope glycoprotein (Env) trimer on the viral membrane is the sole target of bNAbs and the key component of vaccination approaches aimed at eliciting bNAbs. Multimeric presentation of Env on nanoparticles often plays a critical role in these strategies. Here, we will discuss the different aspects of nanoparticles in Env vaccination, including recent insights in immunological processes underlying their perceived advantages, the different nanoparticle platforms and the various immunogenicity studies that employed nanoparticles to improve (neutralizing) antibody responses against Env.

## 1. Introduction

In the early 1980s, human immunodeficiency virus type 1 (HIV-1) was identified as the causative agent of the acquired immune deficiency syndrome (AIDS) epidemic. Since then, an estimated 77.3 million people have been infected by HIV-1 and 35.4 million have died from AIDS-related illnesses. The development of antiretroviral therapy (ART) has helped to significantly reduce the number of AIDS-related deaths (1.9 million in 2004, 0.9 million in 2017) [1]. However, a vaccine will be essential to end the HIV-1/AIDS pandemic [2].

Neutralizing antibodies (NAbs) are the most important correlates of protection for most anti-viral vaccines. The envelope glycoprotein (Env) trimer on the virion surface is the sole target for NAbs against HIV-1. Viral Env is expressed as a gp160 precursor that is proteolytically cleaved into two components: the gp41 transmembrane glycoprotein and the gp120 surface glycoprotein. Together, six molecules of gp41 and gp120 form a trimer of heterodimers. The cleavage process is essential for viral entry as it liberates the fusion peptide, which is necessary for cell membrane attachment and viral fusion [3,4]. The first step in HIV-1 entry is the binding of gp120 on pre-fusion Env to the CD4 receptor on the target cell. The binding of gp120 to CD4 initiates conformational changes that open the recognition site on gp120 for engaging the co-receptor (CXCR4 or CCR5). In turn, the cap that is formed by the trimer of gp120 molecules is opened up, which liberates the underlying metastable gp41 to undergo a loop-to-helix conformational change that drives the fusion peptide into the target cell membrane. These three post-fusion state gp41 molecules form a six-helix bundle, which pulls together the viral and target membrane to facilitate virus infection [5,6].

## 2. Env as a Broadly Neutralizing Antibody Target

HIV-1 has evolved several strategies that protect Env against recognition by NAbs. First, HIV-1 is a rapidly mutating virus, which has resulted in enormous amino acid variation of up to 35% between different Envs [7]. Thus, a NAb that targets one HIV-1 strain often does not recognize another strain. Second, the dense glycan shield surrounding Env acts as a barrier that protects the more immunogenic protein surface from NAb recognition. Strain-specific breaches in the glycan shield that expose the underlying protein surface can be targets for NAbs, but these usually only target the same “glycan hole” on the sequence-matched virus [8,9]. Third, Env is very unstable and conformationally dynamic. The instability of Env causes shedding of the non-covalently attached gp120 subunits, which leaves non-functional gp41 stumps on the viral surface [10]. These stumps and floating gp120 subunits present epitopes that are not present on functional infectious Env trimers. The dynamic structure of Env samples different conformations, which expose non-neutralizing or transient epitopes that are not useful for NAbs. It has been suggested that the instability and dynamic nature of Env are additional strategies of the virus aimed at diverting NAb responses [11].

Despite these hurdles, we know that 20–30% of infected individuals develop broadly neutralizing antibodies (bNAbs) that can neutralize up to 98% of all viral strains. Hundreds of these bNAbs have now been isolated and these target different epitopes that together cover the entire Env surface [12]. Numerous passive immunization studies have shown that bNAbs protect rhesus macaques against simian-human immunodeficiency virus (SHIV) infection and a vaccine able to induce bNAbs will likely protect humans against HIV-1 infection as well [13]. Therefore, many experimental HIV-1 vaccines are now focused on eliciting bNAbs.

All vaccines designed to induce HIV-1 bNAbs employ Env immunogen(s) in some shape or form. Discussing the different Env configurations that can be used as immunogens, such as peptides and gp120 subunits, is beyond the scope of this review and have been summarized elsewhere [14]. However, of particular relevance are soluble native-like Env trimers, such as the so-called “SOSIP” trimers, that closely mimic the conformation of pre-fusion Env on the virus [15]. Soluble native-like Env trimer antigens do present bNAb epitopes, but not the undesirable, and potentially distracting, non-neutralizing epitopes that are presented by post-fusion Env [15,16,17]. Importantly, native-like Env trimers are capable of inducing NAbs against circulating, more neutralization-resistant (“Tier-2”) viruses [18]. In contrast, non-native Env trimers (“pseudotrimers”) or monomeric gp120 designs usually do not induce Tier-2 NAbs, probably because these immunogens do not optimally present the desired neutralizing viral epitopes [15,18,19]. However, the Tier-2 NAbs induced by native-like Env trimers usually only neutralize the sequence-matched virus and often target isolate-specific kinks in the glycan shield [8,9,20]. Novel Env antigen designs and immunization strategies are being explored to improve the breadth and potency of these NAb responses and ultimately elicit bNAbs [21].

Effective anti-viral subunit vaccines, such as those against hepatitis B virus and human papillomavirus, employ nanoparticles (NPs) and induce high levels of protective NAbs [22]. This suggests that NPs could also play an important role in vaccine strategies aimed at inducing HIV-1 bNAbs. Here, we will review the immunological benefits of NPs for HIV-1 vaccines. We will discuss the different NP platforms and strategies that are being used, and analyze the NAb responses reported by studies that have employed NPs presenting Env trimers.

## 3. Nanoparticles and Immunity

Most viral surfaces consist of an outer shell or envelope that presents several (different) proteins in a highly repetitive manner. The immune system has evolved to recognize such repetitive structures [23]. However, the HIV-1 virion only carries 10–20 Envs and it has been hypothesized that this low number is another strategy of HIV-1 to avoid immune recognition [24]. This property can be overcome by using NPs that present Envs in a highly repetitive manner. A number of immunological mechanisms might explain why presenting Envs on NP could improve humoral immune responses against HIV-1.

Highly repetitive surface patterns can initiate B cell receptor (BCR) cross-linking and deliver a strong activation signal to the B cells [25]. These highly repetitive patterns are emulated on the surface of NPs. A scenario in which multiple binding events can occur between the host cell BCRs and the NP, opposed to monovalent binding of single soluble antigens, can be critical in the induction of a potent immune response.

In order to survive the B cell compartments in germinal centers, B cells compete to accrue antigen, which depends on the affinity of the B cell receptors for the antigen [26]. NP presentation of Envs augments the avidity of potential B cell epitopes, which increases the chance that the cognate B cells recognize those epitopes and bind Env antigens. This is especially important when targeting rare naive B cells that recognize subdominant broadly neutralizing epitopes [27]. Abbott et al. demonstrated that precursor (germline) B cell frequency, antigen affinity, and avidity can independently be limiting factors for the competitive success of bNAb precursor-expressing B cells, in this case the precursor B cells of the CD4 binding site bNAb VRC01. They showed that germline-VRC01-targeting NP immunogens with high affinity were able to stimulate the rare VRC01-class B cells to compete efficiently with other B cells in the germinal centers for the antigen. Antigen multimerization had a large impact on the interclonal competitive fitness of the specific VRC01-class B cells.

Transport of antigens to B cells in the lymph nodes is different for small and large antigens [25]. Whereas low-molecular-weight antigen transport is mediated by conduits and directly reach the B cell region [28], larger antigens are bound by subcapsular sinus macrophages in the form of immune complexes (ICs). Blood-circulating low-affinity immunoglobulin M (IgM) can bind to a pathogen if it contains suitable epitopes that are repetitively presented and organize into ICs, sometimes combined with components of the complement system. The ICs are subsequently transferred to follicular dendritic cells (FDCs) in the B cell follicles [25]. It was recently shown that rapid trafficking of Env-NPs to FDCs is mediated by mannose-binding lectin (MBL) that usually binds to glycosylated microbes and activates complement via MBL-associated serine proteases. Furthermore, it was shown that the dense array of glycans on NPs coated with highly glycosylated Env trimers triggered MBL-mediated innate immune recognition, but soluble Env trimers did not [29].

When designing Env–NPs for vaccines one needs to consider these underlying immunological mechanisms to improve immunogenicity. The size of NPs is a critical element in its interaction with the immune system [30]. Subcutaneously injected smaller NPs (20–200 nm) traffic more efficiently to draining lymph nodes than larger particles (500–2000 nm), because small particles have the capacity to freely drain to the lymph while large NPs rely on the migration of dendritic cells [31]. Larger particles are substantially more retained in the lymph node tissue [32]. Particle size also affects the uptake of NPs by antigen presenting cells. From a set of 17, 7.0, 3.0 and 0.3 µm diameter poly (lactic-co-glycolic acid) (PLGA) particles, the smallest 0.3 µm NPs were most efficiently taken up by dendritic cells in vitro [33]. This is in agreement with earlier work on NPs with complement activating surfaces, which demonstrated that 50% of 25 nm NPs were transported to dendritic cells in the lymph nodes, whereas 100 nm particles were only 10% efficient [34].

Optimal spacing and high density of epitopes are important for strong activation of B cells. It has been shown that a spacing of 5–10 nm of 15–20 hapten molecules is ideal for B cell activation [35], which is similar to the average spacing of most spike proteins on viral surfaces [36]. A classic immunization study demonstrated that more than 12–16 appropriately spaced hapten groups are necessary for a proper immune response [37]. A more recent study demonstrated that NPs displaying a higher density of Env variable loop 3 peptides induced higher binding Ab titers [38]. Similarly, high-density display of respiratory syncytial virus (RSV) prefusion F glycoproteins improved RSV NAb responses compared to low-density display of F proteins using the same NP platform [39].

## 4. Nanoparticle Platforms

### 4.1. Virus-Like Particles

Virus like particles (VLPs) are NPs that closely resemble the original virus and are usually produced by expressing one or more viral proteins. However, these particles lack the genetic information of the virus [40]. HIV-1 virions are ~110 nm in diameter and display approximately 10–20 Env trimers on their surface in an irregular distribution [41]. The relatively large size and low-density distribution of Env trimers make VLPs based on existing HIV-1 virions an undesirable platform for antigen-presentation. Furthermore, HIV-1 virions also display non-functional forms of Env that might distract humoral responses [10]. The latter issue can be solved by eliminating the non-functional Env forms using enzymatic digestion [42]. The Env trimers on these particles display essentially all bNAb epitopes in the context of the lipid membrane, which also forces the correct angles of approach for antibodies. Rabbits immunized with these “trimer VLPs” induced Tier-2 neutralization [20,43]. However, the irregular distribution and low density of Env trimers on these VLPs remain a disadvantage compared to other platforms. Furthermore, producing the trimer VLPs is very inefficient: Crooks et al. estimated that the yields of SOSIP antigen are >100 fold higher compared to VLPs when using an equivalent expression system [43]. In an attempt to generate HIV-1 VLPs with increased Env density, human cells were transduced with native Env and sorted using fluorescence-activated cell sorting (FACS) for a phenotype featuring high binding to bNAbs and low level of non-NAb binding [44]. The VLPs generated using this method presented an average of >120 Env trimers per virion. How these “high-Env” VLPs perform in vivo remains to be addressed.

### 4.2. Liposomes

Liposomes have become important carriers in vaccine formulations. The development of two commercial liposomal vaccines, Inflexal^®^ V and Epaxal^®^, against influenza and hepatitis A virus, respectively, have increased the interest in these vesicles in vaccine research. Liposomes are spherical vesicles made of a lipid bilayer with an aqueous cavity, which can be used to present antigens on the surface or encapsulate them [45]. Liposomes are an attractive and versatile platform for presenting a number of different Env immunogens.

The membrane proximal external region (MPER) of Env gp41 is a target for a number of bNAbs and the MPER is (partly) embedded in the viral membrane [46,47,48]. Several MPER bNAbs also directly interact with the adjacent lipid bilayer; e.g., 2F5 [47] and 4E10 [48]. Embedding MPER peptides on liposomes can improve their antigenicity, since this resembles the presentation of the MPER on the native virus more closely. Indeed, MPER-liposomes are able to elicited binding Abs that target epitopes that overlap with those of anti-MPER bNAbs [49]. In contrast, adjuvanted soluble MPER peptides not anchored in liposomes hardly elicited MPER-specific antibody responses [50]. However, the binding Abs induced by MPER-liposomes usually do not have neutralizing activity, possibly because these immunogens induce Abs that have angles of approach that are not compatible with those on viral Env. Thus, while liposomes greatly improve the immunogenicity of MPER peptides, additional parameters probably need to be optimized for inducing HIV-1 NAbs using MPER-liposomes.

Liposomes can also be used for presenting repetitive arrays of complete Env trimers. Interbilayer-crosslinked multilamellar vesicles (ICMVs) are highly stable liposomes that are suitable for stably entrapping and displaying antigens for inducing potent humoral and cellular immune responses [51,52]. Non-native uncleaved Env trimers anchored to ICMVs via a non-covalent interaction between the polyhistidine-tags on the Env trimers and nickel-nitrilotriacetic acid (Ni-NTA)-headgroup lipids, induced ~50-fold higher Ab binding titers in mice compared to soluble trimers. Also, immunizations with the ICMVs increased the number of epitopes recognized by the binding Abs [53]. In another study, germline targeting SOSIP trimers were linked to conventional liposomes with the same his-tag/nickel conjugation. Liposomes could activate PGT121 germline B cells at concentrations 1000-fold lower than the soluble germline targeting trimers in vitro [54]. Furthermore, mice immunized with liposomes bearing Env trimers showed increased binding Ab titers and increased germinal center activation compared to mice immunized with soluble Env trimers [55,56,57]. However, none of these immunized wild type mice elicited Abs that were capable of neutralizing Tier-2 viruses, probably because murine B cells usually do not elicit such NAbs [58].

Rabbits, guinea pigs and non-human primates are able to induce Tier-2 NAbs. Rabbits immunized with liposomes carrying clade B JRFL SOSIP trimers induced marginally improved binding Ab and NAb titers compared to rabbits immunized with soluble SOSIP trimers. However, these findings were not statistically significant [56]. In another study, rhesus macaques were immunized with liposomes displaying an array of clade C 16055 native flexibly linked (NFL)-Env trimers and the serum of these animals showed significantly improved NAb activity against the autologous Tier-2 virus compared to soluble 16055 NFL-Env trimers. Moreover, superior germinal center responses were observed for the liposome immunized animals [59]. Notable is that several studies reported the instability of the his-tag/nickel conjugation at increased temperatures or in serum. Therefore, liposomes were developed that contain lipids with maleimide head-groups that covalently couple to C-terminal cysteines on engineered Env trimers [54,56,59]. However, rhesus macaques immunized with liposomes displaying covalently coupled BG505 Env trimers did not show improved autologous neutralization compared to soluble trimers [60] (see also Section 5). In summary, liposomes are a versatile platform for immunogen presentation, and can also carry adjuvants for improved immunogenicity. However, antigen coupling efficiency, size, particle integrity and manufacturing consistency remain issues that need to be addressed.

### 4.3. Self-Assembling Protein Nanoparticles

Many naturally occurring proteins can self-assemble into symmetrical and stable NPs and some of these are suitable for antigen presentation [61]. Next to naturally occurring proteins, a number of de novo self-assembling protein NPs have been engineered using computational design [62,63,64], including, ultra-stable artificial protein nanocages displaying reversible assembly [65]. A big advantage of protein NPs is their suitability for large-scale production via bacterial and, highly relevant for HIV-1 Env NP immunogens, eukaryotic expression systems.

The self-assembling ferritin nanocage consists of 24 subunits. Its octahedral symmetry is exceptionally suitable for displaying a variety of antigens, such as hemagglutinin (HA) of influenza virus [66], E2 of hepatitis C virus [67,68], gp350 of Epstein Barr virus [69] and HIV-1 Env, including gp120 monomers and complete Env trimers [29,70,71,72,73,74].

Mice and rabbits immunized with ferritin NPs presenting native-like BG505 SOSIP trimers showed higher trimer-binding Ab titers compared to animals immunized with soluble trimers [29,71,72]. Surprisingly, the autologous 50% neutralization titers against the autologous Tier-2 BG505 virus were not improved in the ferritin-immunized rabbits [71,72]. In contrast, rabbits immunized with ferritin NPs carrying the consensus sequence-based ConM SOSIP trimer showed increased neutralization titers compared to rabbits immunized with soluble ConM SOSIP trimers [74]. This apparent contradiction is discussed in the next paragraph.

The ferritin NPs mentioned above consist of 24 copies of the same subunit, but other ferritin NPs consist of twelve copies of two different subunits that also self-assemble into a 24-mer. These ferritin NPs have been used for presenting HIV-1 and influenza antigens on the same NP and when administered to guinea pigs elicited NAbs against Tier-1 HIV-1 and influenza viruses [70]. These dual-display ferritin NPs open new possibilities for novel strategies aimed at inducing NAbs against HIV-1, but also other viruses.

However, the octahedral symmetry of ferritin restricts the maximum number of displayed antigens to 24 monomers or eight Env trimers, which might limit their immunogenicity compared to other NP platforms. Thus, other self-assembling NPs, such as the dihydrolipoyl acetyltransferase (E2p) from *Bacillus stearothermophilus* and the computationally designed I3-01 NP that can harbor up to 20 Env trimers could be more promising platforms. Indeed, both NPs activate B cells efficiently in vitro, but their performance as immunogens in vivo have not been tested yet [71,73].

The self-assembling protein NP lumazine synthase from *Aquifex aeolicus* has been used for multimeric presentation of the engineered outer domain (eOD) of gp120. The eOD-60mer engages (predicted germline) bNAbs of the VRC01-class more strongly than the eOD monomer or trimer in vitro [75]. Moreover, the 60-mer form of eOD induced stronger B cell responses than eOD trimers in knock-in mice transgenic for the germline-reverted VRC01 heavy chain [76]. Promisingly, the eOD proteins can be used to isolate VRC01-class precursor naive B cells in HIV-uninfected donors [75,76,77]. The eOD 60mer NP is now being evaluated as an immunogen in a phase I clinical trial (clinicaltrial.gov: NCT03547245).

Ferritin, E2p, I3-01 and lumazine synthase are Env-carrying NPs that are assembled intracellularly, and this might be suboptimal, because it is difficult to ensure that all Envs on the NPs are well-folded. Indeed, a number of ConM SOSIP trimers on ferritin NP were uncleaved and presented non-neutralizing epitopes that induce unwanted V3-specific non-NAbs [74]. Two-component platforms that allow in vitro assembly of NPs, allow for better control over Env quality, as the Env trimers are purified prior to assembly [78]. For example, the bacteriophage AP205 VLPs carrying the SpyCatcher protein on their exterior can be used to covalently couple separately purified SpyTagged SOSIP trimers [79,80]. Another promising platform are the I53-50NPs that consist of two components: twenty I53-50A trimeric subunits and twelve I53-50B pentameric subunits. The I53-50NP assembles by simply mixing these two components. These artificial computationally designed NPs can display up to 20 trimeric antigens [62]. I53-50NPs with 20 RSV F trimers induced ~10-fold stronger NAb responses in mice and macaques than soluble RSV F trimers [39]. In our lab, the I53-50NP with ConM SOSIP trimers elicited over 40-fold stronger autologous neutralization than soluble ConM SOSIP trimers and also outperformed ConM SOSIP-ferritin NPs in eliciting autologous NAb responses after the first immunization [81]. Furthermore, these and similar two-component platforms allow for mixing of multiple Env trimers on the same NP (discussed later).

## 5. Accessibility of Neutralizing Epitopes on Env Nanoparticles

The accessibility of (neutralizing) epitopes changes when soluble Env trimers are attached to NPs. The biggest proteinaceous area of the soluble Env trimer is the bottom [58]. However, the Env trimer bottom is usually hidden by the membrane on HIV-1 virions, but it is a highly exposed neo-epitope on soluble trimers that induces immunodominant non-NAbs [58,82]. NP presentation of Env occludes this non-neutralizing bottom epitope.

On the other hand, it is possible that NP presentation restricts access of certain neutralizing epitopes to B cell receptors. For instance, the most immunodominant neutralizing epitope on BG505 Env is the 241/289 glycan hole near the base of the trimer [8]. Base-proximal (bNAb) epitopes, like the gp120/gp41 interface and gp41 epitopes, are usually less efficiently presented on ferritin and I53-50 NPs (and possibly also other NP platforms) and might therefore be less immunogenic in vivo [71,72,81]. This might explain why BG505 Env trimers on NPs, including ferritin [71,72], liposomes [60] and I53-50NPs [81] did not seem to induce improved NAb responses compared to soluble BG505 Env trimers. In contrast, epitopes located at or around the trimer apex are more efficiently presented on NPs than on soluble trimers, probably due to the accessibility of apex epitopes on NPs and the increased epitope avidity provided by the NP. The 16055, JRFL and ConM Envs have immunodominant neutralizing epitopes near the trimer apex and the individual studies reported a trend or a significant improvement in the NAb response for animals immunized with NPs carrying Envs with an immunodominant apex [20,59,74,81].

To determine the effect of NP presentation on the induction of HIV-1 NAbs, we performed a meta-analysis on the autologous NAb titers reported in immunogenicity studies that compared the aforementioned native-like Env trimers on NPs to their soluble counterparts in rabbits and non-human primates (Figure 1). When all studies were combined, we found that NAb titers were ~2.4-fold higher for animals immunized with NPs compared to animals immunized with soluble trimers (*p* = 0.0026; *n* = 105 animals; Figure 1A). However, we did not find such an improvement for NPs when we compared the autologous NAb titers of BG505 Env immunized animals only (*p =* 0.9961; *n* = 50 animals; Figure 1B, left graph). In stark contrast, we found that NPs induced a significant ~5.2-fold higher NAb titer when only Envs with immunodominant NAb epitopes on the trimer apex (i.e., 16055, JRLF and ConM Envs) were taken into account (*p* < 0.0001; *n* = 55 animals; Figure 1B, right graph). These analyses suggest that the accessibility of the neutralizing epitope(s) determines whether NPs are useful for increasing the immunogenicity of Env trimers.

Additional studies are needed to determine if the BG505 NAbs induced by NPs are redirected to other, possibly less immunogenic, epitopes on BG505 Env [83,84]. However, if targeting of base-proximal epitopes such as the 241/289 glycan on BG505 Env would be desired, one could change Env density or the design of the NP. For example, by presenting the trimer further away from the NP cage the accessibility of base epitopes could be increased.

However, the ultimate goal of many HIV-1 vaccines is to induce bNAbs, and the 241/289 glycan hole seems highly specific for BG505 Env and might not be a preferable target for initializing a bNAb response. A better strategy would be to target responses to more conserved epitopes, such as the CD4 binding site.

## 6. Epitope Focusing Using Nanoparticles

The high number of somatic mutations and expansive rounds of selection in the germinal centers needed for developing bNAbs, suggest that it is necessary to activate the B cells that express the appropriate germline bNAb precursor. Focusing the response to these broadly neutralizing epitopes is one of the main challenges for developing an HIV-1 vaccine. However, most Envs are not recognized by the naive (germline) B cells that have the capacity to develop into bNAb-producing B cells [85,86]. Therefore, immunogens are being developed that recognize inferred germline bNAbs [54,75,87] and employed in strategies that would prime the right germline B cell and guide these B cells towards breadth [76,88,89,90].

In many of these germline-targeting strategies, NPs are essential as a priming immunogen for starting the right germline bNAb, possibly because the affinity of naive B cells for these engineered immunogens is still relatively low. Several examples highlight the importance of nanoparticles for priming germline bNAb responses in vivo. The eOD-60mer more efficiently activated VRC01-class B cells in transgenic mice than the eOD-trimer [76]. Germline-3BNC60 targeting gp120 monomers fused to ferritin NPs were able to efficiently activate the cognate B cells in transgenic mice, while soluble trimeric forms did not [91]. The B cells expressing germline 3BNC60 were autoreactive and this might explain why the multimerized NP form induced increased Ab responses in the transgenic mice [91]. In a recent study, a germline targeting native-like Env trimer derived from the BG505 SOSIP, called RC1, was designed to recruit V3-glycan-specific B cells by improving the accessibility of the V3-glycan patch epitope. Immunizations with RC1 activated B cells expressing Abs that resemble human V3-glycan patch bNAb precursors in mice, rabbits and macaques [82]. Future studies are needed to determine whether the responses induced in these animal models can be guided towards breadth.

Another interesting epitope focusing strategy is using heterotypic NPs that display different Env immunogens on the same NP surface. In theory, this would diminish the activation of B cells that recognize strain-specific epitopes and allow for selective engagement of B cells that recognize conserved cross-reactive epitopes shared by the different Envs. This concept has been proven to work for influenza H1N1 [92]. In that study, the receptor binding domains of HA spikes from eight different H1N1 viruses were presented on a single ferritin NP. These heterotypic HA–NPs elicited B cell responses that were quantitatively and qualitatively superior to those elicited by monovalent HA–NPs or a mix of eight homotypic HA–NPs [92]. Obviously, similar strategies could be employed for HIV-1 vaccines as well. Several points need to be considered: will the heterotypic Env–NPs be utilized for targeting a particular epitope or the complete Env surface? How diverse should these Envs: NPs with highly diverse Envs might have a lower chance of success, but might induce broader responses, while NPs that would carry more closely related Envs possibly have an increased chance of generating a potent NAb response. But it is clear that the heterotypic NP strategy is a very promising research avenue for HIV-1 vaccinology.

## 7. Anti-NP Responses

It is inevitable that the exposed NP surface that is accessible to the immune system will trigger humoral responses. Indeed, ferritin and I53-50 NPs elicit binding Ab responses [39,66,74], but these Abs did not interfere with the induction of NAbs in the case of RSV and influenza virus. However, anti-NP responses might interfere with the induction of HIV-1 NAbs, because Env is not very immunogenic. Similarly, the neo-epitope on the bottom of soluble Env trimers is highly immunogenic and is the initial target of most Abs after the soluble Env trimer immunization [83]. NP presentation of Env could be a double-edged sword because of the immunogenic surface of NPs: anti-Env responses are improved because of the repetitive display of Env epitopes on NPs, but unwanted anti-NP responses potentially limit the strength of the desired anti-Env responses. None of the aforementioned HIV-1 NP studies properly addresses this potential issue. However, it is well-documented that pre-existing anti-vector immunity lowers the immunogenicity of adenovirus vector-based vaccines [93]. This pre-existing anti-vector immunity can be circumvented by changing the adenovirus vector. For example, by replacing immunogenic regions of the hexon protein by the corresponding region from a rare adenovirus serotype. Pre-exiting vector immunity abrogated the desired immunogenicity of the parental adenovirus vectors, whereas the immunogenicity of the chimeric vector was not suppressed [91]. Importantly, it has been suggested that pre-existing anti-vector immunity was the cause for the failure of the Step Study [94]. Considering these potential negative effects, one could consider decreasing unwanted anti-NP responses by using less immunogenic NP platforms, such as liposomes, or by covering the immunogenic parts of NPs with polyethylene glycol [52] or glycans [95].

## 8. Conclusions

In recent years, the interest in NPs for HIV-1 subunit vaccines has drastically increased. New NP platforms are now being explored to improve immunization strategies to induce HIV-1 NAb responses. Furthermore, there is increasingly more known about the immunological mechanisms underlying the immunogenicity of NPs. Using a meta-analysis, we have demonstrated that NPs boost autologous HIV-1 NAb responses, but that this effect is probably dependent on the accessibility of the NAb epitope. Thus, knowing the desired NAb epitope will help to optimize the presentation of certain Envs and improve their immunogenicity on NPs. Furthermore, NPs will likely be important as priming immunogen for targeting low affinity naive B cells through the high avidity interactions provided by the NP display. A successful HIV-1 vaccine will likely employ NPs for optimal immunogenicity, but additional research is necessary to exploit them to their full potential.

## Figures and Tables

**Figure 1 vaccines-07-00076-f001:**
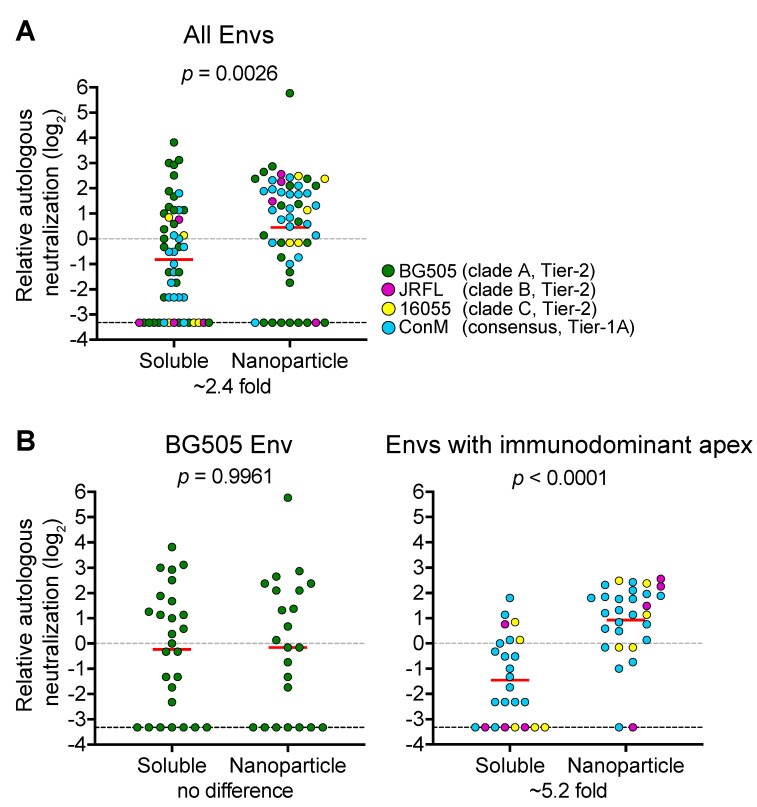
A meta-analysis comparing neutralizing responses induced by native-like Env trimers on nanoparticles or as soluble immunogens. The autologous neutralization data from rabbits and non-human primates were taken from the following studies: for BG505: [60,71,72,81]; for JRFL: [56]; for 16055 [59]; for ConM [74,81]. To avoid bias caused by differences in immunogenicity between these Envs, we first determined the geometric mean for each study at time of peak neutralization titer (i.e., two weeks after the last immunization). Subsequently, the relative neutralization titers of the individual sera were calculated using the geometric mean. When no neutralization was detected (i.e., 50% neutralization titer of <20), the value of this animal serum was set to 0.1. (**A**) NAb titers from animals immunized with BG505, JRFL, 16055 and ConM NPs or soluble Env. (**B**) NAb titers from animals only immunized with BG505 NPs or soluble Env (left); NAb titers from animals immunized with JRFL, 16055 and ConM NPs or soluble Env (right). Fold differences were calculated by comparing geometric means (denoted by horizontal red lines) between the two data sets. A Mann–Whitney U test was used to determine significance.

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
