# Peer review of "Nanoparticle Vaccines for Inducing HIV-1 Neutralizing Antibodies"

_vaccines, 2019, doi:10.3390/vaccines7030076_

Round 1

Reviewer 1 Report

I thank you the authors to prepare the manuscript. 

The following lines in the text have either have grammatical errors or the concept is not clearly given; 28, 73, 120, 123-128, 169-174, 203-205, 240, 244, 245, 247, 248, 256-258, 261, 300, 

Author Response

We thank you the authors to prepare the manuscript. 
The following lines in the text have either have grammatical errors or the concept is not clearly given; 28, 73, 120, 123-128, 169-174, 203-205, 240, 244, 245, 247, 248, 256-258, 261, 300, 

We thank the referee for detailed reviewing of our manuscript. We have altered the text in the identified sections.

Reviewer 2 Report

The review by Brinkkemper and Sliepen covers the application of nanoparticle vaccines for HIV vaccine development. The literature cited is up to date and highly relevant, and the text is clear and well written.  Comments below:

1 – The immunogenicity boost from nanoparticle-based vaccines is well reported in the literature and nicely summarized in this review. However for HIV, it could be argued that the primary vaccine challenge has never been immunogenicity per se (the failed Vaxgen, RV144 etc vaccines all elicited good antibody titres.)  Instead, the challenge has been epitope focusing the antibody response onto broadly neutralising targets.  Although the authors touch briefly on this topic, I feel the review would be strengthened by further expanding the discussion around the potential for NP vaccines to facilitate epitope focusing, examples in the literature where this has worked/failed and the potential caveats such as NP occlusion of valuable epitopes.

2 – The section around anti-NP responses is very short.  Many groups have measured anti-vector responses, what actual impact do these responses have on re-immunisation?

Minor:

Line 282 – typo – “presenting the f the trimer”

Author Response

The review by Brinkkemper and Sliepen covers the application of nanoparticle vaccines for HIV vaccine development. The literature cited is up to date and highly relevant, and the text is clear and well written.  Comments below:

We thank referee #2 for his or her positive remarks regarding our review paper. We would like to note that we have changed the text throughout the manuscript to further improve readability.

1 – The immunogenicity boost from nanoparticle-based vaccines is well reported in the literature and nicely summarized in this review. However for HIV, it could be argued that the primary vaccine challenge has never been immunogenicity per se (the failed Vaxgen, RV144 etc vaccines all elicited good antibody titres.)  Instead, the challenge has been epitope focusing the antibody response onto broadly neutralising targets.  Although the authors touch briefly on this topic, I feel the review would be strengthened by further expanding the discussion around the potential for NP vaccines to facilitate epitope focusing, examples in the literature where this has worked/failed and the potential caveats such as NP occlusion of valuable epitopes.

We thank the reviewer for bringing up these important points. The benefits of NP presentation of Env immunogens is evident, and we have tried to highlight this by summarizing the available in vivo neutralization data in Figure 1A.
We agree with the reviewer that NPs can be used to steer responses towards desired (germline) bNAb epitopes and we now added and rewritten sections of paragraph 6 to further elucidate this topic.
The referee rightly remarks that NP presentation can actually hide certain desired NAb epitopes and this is something that is usually not addressed in studies employing Env-NPs. We tried to convey this point by showing in Figure 1B that in the case of BG505 Env there is no benefit of NP presentation, possibly because the immunodominant 241/289 glycan hole is occluded on NPs. The latter has now been clarified more in section 5 (
lines 529-670). We have also added a sentence about the decreased binding of certain base-targeting or gp41/gp120 interface bNAbs (e.g. 35O22, PGT151) (lines 530-534)

2 – The section around anti-NP responses is very short.  Many groups have measured anti-vector responses, what actual impact do these responses have on re-immunisation?

Not much is known about the influence of anti-NP responses on Env vaccinations aimed at NAbs. However, it is known that pre-existing immunity against a delivery vector might decrease the desired immunogenicity of the vaccines or have even been implicated with harm in the case of the Step Trial. We have now added text and the relevant references (lines 916-917). We have now also added a section on the influence of pre-existing anti-NP responses in the context of two different pre-clinical vaccines studies (lines 910-920).

Minor:

Line 282 – typo – “presenting the f the trimer” 

We have now changed this.